# Characteristics and trends of medical malpractice claims in Japan between 2006 and 2021

Kaori Taniguchi[1], Takashi Watari[2,3], Kiwamu Nagoshi[1] *

1 Department of Environmental Medicine and Public Health, Shimane University, Izumo, Shimane, Japan,
2 General Medicine Center, Shimane University Hospital, Izumo, Shimane, Japan, 3 Department of
Medicine, University of Michigan Medical School, Ann Arbor, MI, United States of America

* nagoshi@med.shimane-u.ac.jp

Characteristics and trends of medical malpractice
claims in Japan between 2006 and 2021. PLoS
ONE 18(12): e0296155. https://doi.org/10.1371/
journal.pone.0296155

Universita degli Studi di Foggia, ITALY

**Data Availability Statement:** The data supporting
this study's findings are available from the Public
Relations Section, Supreme Court of Japan. Data
are available from the institution of corresponding
author (Department of Environmental Medicine &

## Abstract

Classification and analysis of existing data on medical malpractice lawsuits are useful in
identifying the root causes of medical errors and considering measures to prevent recur-
rence. No study has shown the actual prevalence of all closed malpractice claims in Japan,
including the number of cases and their trial results. In this study, we illustrated the recent
trends of closed malpractice claims by medical specialty, the effects of the acceptance rates
and the settlements and clarified the trends and characteristics. This was a descriptive
study of all closed malpractice claims data from the Supreme Court in Japan from 2006–
2021. Trends and the characteristics in closed malpractice claims by medical specialty and
the outcomes of the claims, including settlements and judgments, were extracted. The total
number of closed medical malpractice claims was 13,340 in 16 years, with a high percent-
age ending in settlement (7,062, 52.9%), and when concluding in judgment (4,734, 35.3%),
the medical profession (3,589, 75.8%) was favored. When compared by medical specialty,
plastic surgery and obstetrics/gynecology were more likely resolved by settlement. By con-
trast, psychiatry cases exhibited a lower likelihood of settlement, and the percentage of
cases resulting in unfavorable outcomes for patients was notably high. Furthermore, there
has been a decline in the number of closed medical malpractice claims in Japan in recent
years compared to the figures observed in 2006. In particular, the number of closed medical
malpractice claims in obstetrics/gynecology and the number of closed medical malpractice
claims per 1,000 physicians decreased significantly compared to other specialties. In con-
clusion, half of the closed malpractice claims were settled, and a low percentage of patients
won their cases. Closed medical malpractice claims in Japan have declined in most medical
specialties since 2006. Additionally, obstetrics/gynecology revealed a significant decrease
since introducing the Obstetrics/Gynecology Medical Compensation System in 2009.

## Introduction

"To Err Is Human," published in 2000, states that it is essential to improve safety by under-
standing medical errors [1]. A previous report estimated that medical error is the third leading

Public Health Faculty of Medicine, Shimane University, e-mail: p-health@med.shimane-u.ac.jp) upon reasonable request and with the permission of the Public Relations Section, Supreme Court of Japan.

**Funding:** The authors received no specific funding for this work.

**Competing interests:** The authors have declared that no competing interests exist.

cause of death (approximately 250,000 deaths per year in the United States of America) after cardiovascular disease and malignant neoplasms [2]. Hence, the causes of such errors need to be identified and resolved at both the individual and system levels to improve patient safety. In addition, medical errors are detrimental to patient health outcomes and a significant burden to healthcare providers and the healthcare economy [3–5].

Medical litigation is a conglomeration of cases in which medical error has likely caused direct patient harm and contains real problems at both the individual and system levels for patients and healthcare providers [6]. Ideally, it is desirable to present incidence and mortality rates resulting from medical errors and conduct a root cause analysis before cases escalate into medical litigation. However, categorizing and analyzing existing data from medical malpractice litigations can also be valuable for identifying the root causes of medical errors and considering measures to prevent their recurrence. This process has the potential to contribute to the improvement of the next generation of healthcare, ensuring higher quality standards. In the United States of America, epidemiological studies of medical litigation have focused on the characteristics and trends of medical litigation by using data from the National Practitioner Data Bank to compare the fluctuations in the number of lawsuits by medical specialty, their judgments, and settlements [6–8]. In contrast, in Japan, an analysis has been conducted to identify and compare errors in high-risk medical specialties, such as internal medicine, neurosurgery, orthopedic surgery, obstetrics/gynecology (OB/GYN), based on detailed clinical information from commercialized malpractice claim case databases [9–13]. However, such databases have a significant selection bias because they include relatively high-profile litigation and do not report all medical litigation. Thus, three unknown variables in previous studies have been acknowledged: 1) the actual characteristics of medical litigation from each medical specialty in all the litigations in Japan; 2) the actual and accurate number and status of judgments, settlements, and lawsuit withdrawals; and 3) changes in medical litigation over time. Therefore, we used the Supreme Court data to compare the recent changes in the number of closed malpractice claims per medical specialty and the results of accepted claims and settlements to identify trends and characteristics.

## Materials and methods

### Study design

We included all closed medical malpractice claims cases in Japan from 2006 to 2021. A descriptive research design was chosen for this study due to the limited number of data points and to avoid bias due to testing. In June 2022, we requested and received permission from the Public Affairs Division of the Supreme Court Administrative Office to provide medical malpractice statistics and to publish the data as "Medical Malpractice Claims Case Statistics."

### Definitions of the claim cases

The national data, furnished by the Japanese Supreme Court, encompassed all verdicts pertaining to closed medical malpractice claims that spanned from the lower courts to claims addressed at the Supreme Court. The data provided in this study consists of aggregated information on the number of medical malpractice claims resolved through legal procedures, along with the litigation outcomes and average trial durations, categorized by medical specialty and year. It does not include specific details such as the reasons for filing the lawsuits, litigation content, or the settlement amounts.

The results of the closed medical malpractice claims provided by the Supreme Court were categorized as "Judgment," "Settlement," "Medical provider admits patient's claim," "Patient waives own claim," "Withdrawal," and "Other." Of these, "Medical provider approves patient's

claim" and " Patient waives own claim" only applied to a few cases and were judged to be a small portion of the total number of cases surveyed; therefore, we included them in the "Other" category. We categorized closed medical malpractice claims into four categories: (a) Judgments, (b) Settlements, (c)Withdrawals, and (d) Others. Judgments were further classified into accepted and rejected claims based on court decisions in favor of the patient or the health care provider. The accepted claims were defined as the court ruling in favor of the patient, while judgments in which the court dismissed patient claims were defined as rejected. A settlement was defined as an agreement to end a court trial before a judgment was reached. The withdrawals category contained cases in which the patient side withdrew from the trial with the consent of the medical provider. To compare the number of physicians in different medical specialties, we used the "Physicians, Dentists, and Pharmacists Data in 2020" [14], a national statistic reported by the Ministry of Health, Labour and Welfare every 2 years. We then categorized the representative specialties into internal medicine, orthopedics, pediatrics, psychiatry, surgery, ophthalmology, OB/GYN, dermatology, otorhinolaryngology, urology, plastic surgery, and others. We determined the percentage of accepted claims, rejected claims, settlements, and withdrawals as primary outcomes. Using each specialty's total number of closed medical malpractice claims per year as the denominator, we calculated the number of closed medical malpractice claims per 1,000 physicians over 16 years. The trial duration refers to the period from when the court accepted the lawsuit until a case concluded with a judgment, settlement, withdrawal, or any other legal resolution.

## Ethics statement

This study was conducted according to the Declaration of Helsinki. Since the data used in this study are publicly available and the statistical data obtained from the Supreme Court do not include comprehensive medical details of individual cases, no approval was required by the Institutional Review Board of Shimane University, and informed consent was not applicable.

## Statistical analyses

We used standard simple descriptive statistics to present closed medical malpractice claims by the medical specialties (internal medicine, surgery, orthopedics, dentistry, OB/GYN, psychiatry, plastic surgery, ophthalmology, urology, pediatrics, otolaryngology, dermatology, anesthesiology, and others). We calculated the settlement, rejected and accepted claims ratio to the denominator of closed medical malpractice claims for each specialty over 16 years. Further, we calculated the number of closed medical malpractice claims per 1,000 physicians by dividing the number of closed medical malpractice claims per medical specialty by the number of physicians in each specialty in the same year as the denominator. All analyses were performed using Excel version 16.16.27 (Microsoft, Office 365; Microsoft Corp., Redmond, WA, United States of America).

## Results

The cumulative number of concluded claims from 2006 to 2021 amounted to 13,340 (Table 1). Of these claims, settlement cases constituted the majority, with a total of 7,062 cases (52.9%). Regarding judgments, out of a total of 4,734 cases (35.3%), 3,589 (75.8%) and 1,145 (24.2%) cases resulted in rejected and accepted claims, respectively. This indicates a higher frequency of dismissed patient claims than judgment claims against practicing physicians. Withdrawal claims constituted 610 cases (4.6%), while the remaining 934 cases (7.0%) were classified as others.

**Table 1. Characteristics of Japanese closed malpractice claims by specialty.**

| Specialty | Judgments | | | Settlements n = 7062 | Withdrawals n = 610 | Others n = 934 | Total closed claims n = 13340 |
|---|---|---|---|---|---|---|---|
| | Rejected claims n = 3589 | Accepted claims n = 1145 | Acceptance rate (%) | | | | |
| **Internal Medicine** | 887 | 258 | 22.5% | 1722 | 139 | 227 | 3233 |
| **Surgery** | 523 | 180 | 25.6% | 1158 | 85 | 179 | 2125 |
| **Orthopedics** | 428 | 119 | 21.8% | 859 | 61 | 99 | 1566 |
| **Dentistry** | 333 | 119 | 26.3% | 678 | 101 | 92 | 1323 |
| **OB/GYN** | 252 | 121 | 32.4% | 658 | 29 | 74 | 1134 |
| **Psychiatry** | 215 | 33 | 13.3% | 159 | 39 | 44 | 490 |
| **Plastic Surgery** | 76 | 44 | 36.7% | 237 | 31 | 19 | 407 |
| **Ophthalmology** | 117 | 21 | 15.2% | 178 | 19 | 26 | 361 |
| **Urology** | 86 | 31 | 26.5% | 137 | 12 | 13 | 279 |
| **Pediatrics** | 66 | 31 | 32.0% | 131 | 9 | 27 | 264 |
| **Otolaryngology** | 60 | 24 | 28.6% | 105 | 6 | 9 | 204 |
| **Dermatology** | 51 | 11 | 17.7% | 99 | 9 | 10 | 180 |
| **Anesthesiology** | 35 | 13 | 27.1% | 47 | 5 | 5 | 105 |
| **Others** | 460 | 140 | 23.3% | 894 | 65 | 110 | 1669 |

Regarding medical specialties, internal medicine had the largest number of closed medical malpractice claims with 3,233 cases (24.2%), followed by surgery (2,125 cases, 15.9%), orthopedics (1,566 cases, 11.7%), dentistry (1,323 cases, 9.9%), and OB/GYN (1,134 cases, 8.5%). The acceptance rate of judgment cases was comparatively high in plastic surgery (44 cases, 36.7%) and OB/GYN (121 cases, 32.4%) and low in psychiatry (33 cases, 13.3%). Similarly, the settlement rate was high in plastic surgery (237 cases, 58.2%) and OB/GYN (658 cases, 58.0%), while the lowest was in psychiatry (159 cases, 32.4%). The distribution of closed medical malpractice claims outcomes by medical specialty is further illustrated in Fig 1.

In 2006, the number of closed medical malpractice claims was at its highest (1,120 cases), although it had significantly declined until 2011 (770 cases). Recently, except for 2020 (674 cases), the number of closed medical malpractice claims has averaged only 700–800 cases per year. By medical specialty, OB/GYN and surgery showed significant reductions in closed medical malpractice claims from 2006 to 2019, with declines from 161 to 44 cases and from 188 cases to 129, respectively. Internal medicine temporarily decreased from 256 to 164 cases between 2006 and 2012 and has been on the rise since reaching 192 in 2019. Plastic surgery and dentistry slightly increased from 20 and 74 cases in 2006 to 35 and 84 cases in 2019, respectively (Fig 2).

In 2006, the rate of cases per 1,000 physicians was 4.9, which decreased to 2.8 in 2020. The figure indicates a decline in the rate of closed medical malpractice claims per 1,000 physicians for most medical specialties. In 2006, the highest rate was noted in OB/GYN (16.8/1,000), followed by plastic surgery (10.5/1000), surgery (8.7/1,000), and orthopedic surgery (6.3/1,000). Nonetheless, the closed medical malpractice claim rate for OB/GYN substantially decreased (3.4/1,000 in 2020), with plastic surgery having the highest closed medical malpractice claim rate (10.7/1,000) in 2020 (Fig 3).

The average length of the trial period for all medical lawsuits in Japan was 24.5 months (all specialties). The longest trials were in pediatrics (33.6 months), anesthesiology (32.9 months), and OB/GYN (29.1 months); the shortest trials were in plastic surgery (18.6 months) and dentistry (18.7 months) (Fig 4).

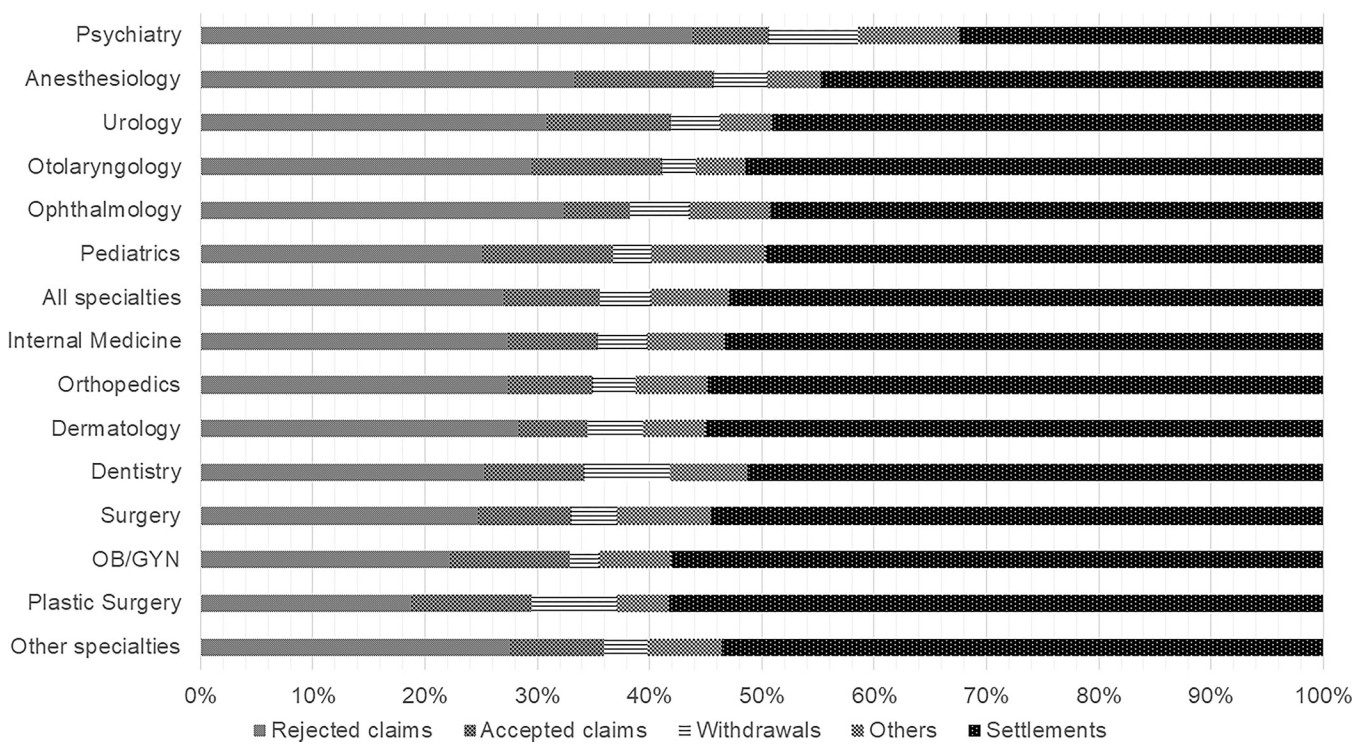

**Fig 1. Litigation conclusions among the medical specialties.**

## Discussion

This descriptive study compiled regular data from 13,340 cases of the Supreme Court of Japan over 16 years to highlight the trends and characteristics of closed medical malpractice claims. Our analysis revealed two key trends. First, the number of closed medical malpractice claims has decreased in most medical specialties since 2006, though there were some specialty-wise differences. Second, most closed medical malpractice claims in Japan were resolved through settlements and judgments.

Our results revealed only 13,340 closed medical malpractice claims in Japan between 2006 and 2021. This is in stark contrast to the estimated 17,000 or more medical malpractice lawsuits that are filed in the United States of America each year, including 75% of the physicians in low-risk specialties and 99% of the physicians in high-risk specialties facing medical malpractice claims at some point in their careers [15]. In addition, defensive medicine practices are generally more common in nations with a higher density of lawyers and tort lawsuits, such as Italy [16]. The lower prevalence of medical malpractice suits in Japan than in other countries may be due to the difference in the judicial system, an economic structure that offers little financial benefit to the patient, and a cultural background that does not favor suing medical personnel [11]. Closed medical malpractice claims during the study period peaked in 2006 and have been fluctuating between 700–800 cases annually in recent years. However, the decrease in the number of medical lawsuits probably does not translate into a decrease in the frequency of patient complaints and dissatisfaction with medical care. Medical ADRs were established in 2007 and are now located in 11 locations nationwide; thus, the number of cases leading to lawsuits may have been suppressed as disputes and are now handled outside of court. As for the reasons courts have ruled in favor over a long period of time, the study was limited to numerical information, and it was difficult to establish a causal relationship. However, prior literature

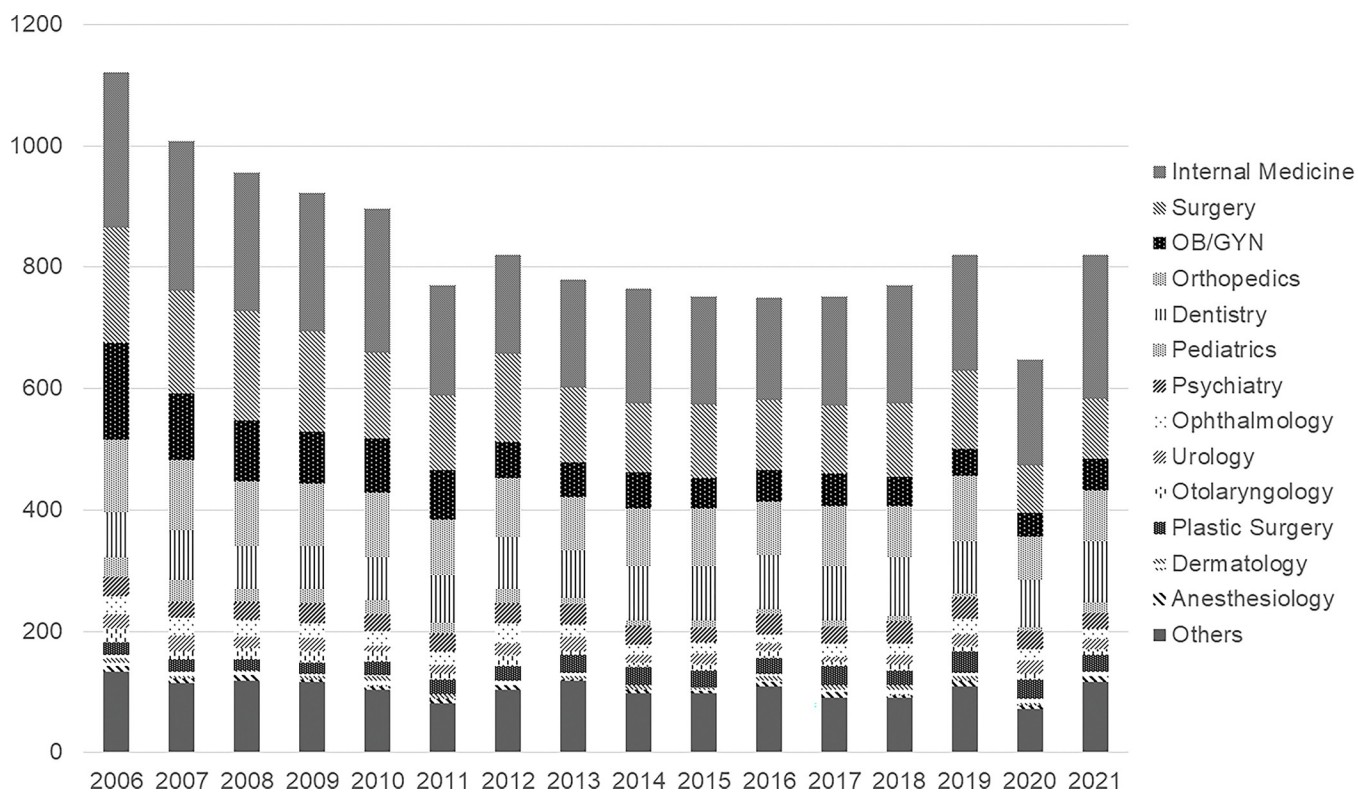

**Fig 2. Number of closed malpractice claims for each specialty.**

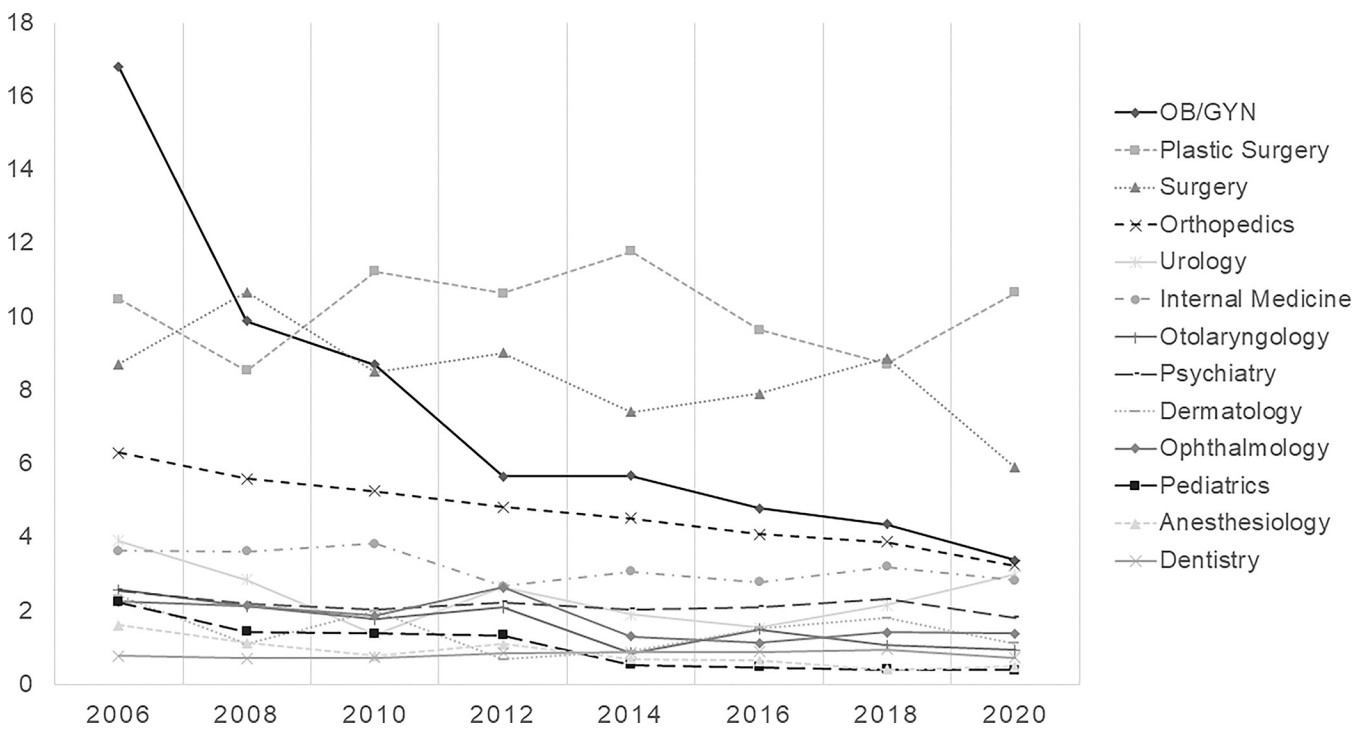

**Fig 3. Trend for the number of closed medical malpractice claims per 1,000 per physicians.**

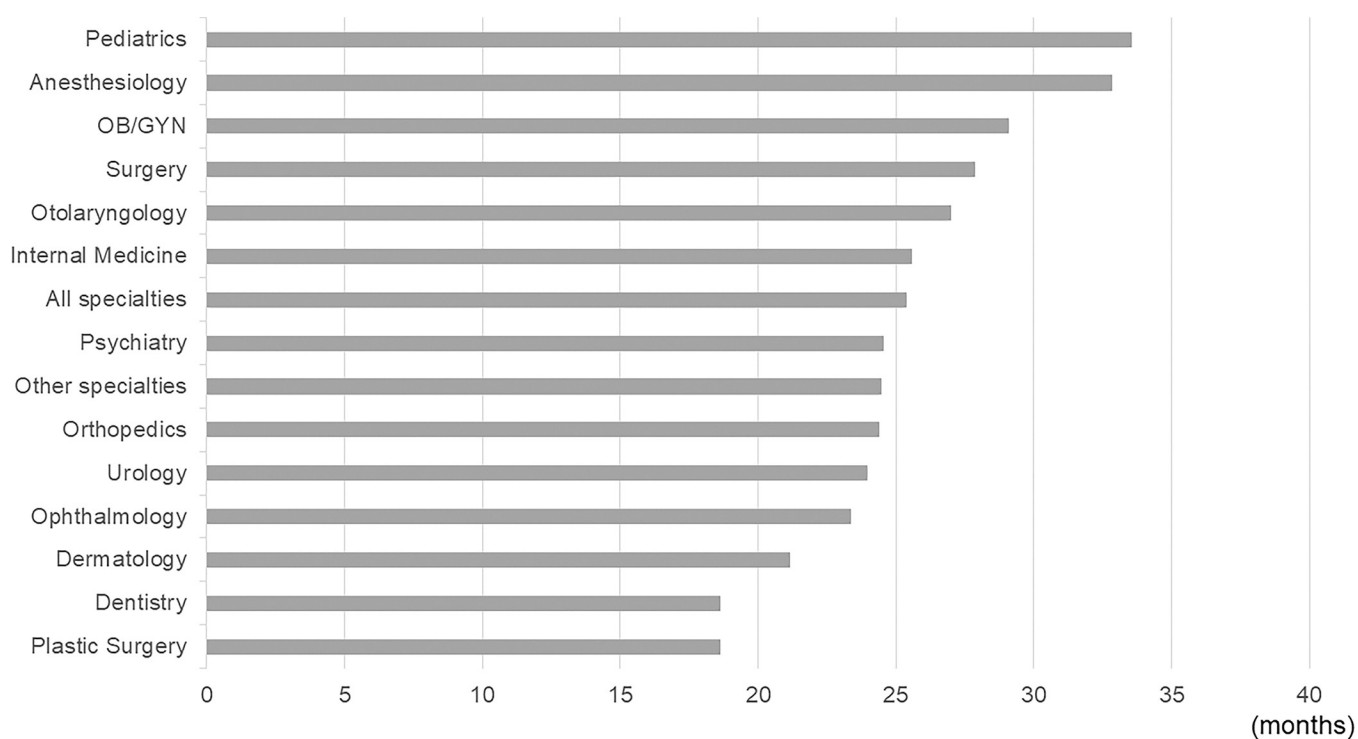

\* Includes judgments, settlements, withdrawals, and any other proceedings that were concluded in court.

**Fig 4. Trial period lengths for each medical specialty from the start of a trial to its conclusion.**

suggests that high transparency in social contexts may have contributed to adopting a broader concept of informed consent and led to stricter judgments about deficiencies in informed consent [17]. We can speculate that around 2006, when medical litigation substantially increased, the sentences against healthcare professionals were somewhat stricter than they are today.

As far as we know, no evidence suggests that the Supreme Court limits the number of medical malpractice cases. However, it is important to note that Japan's legal system allows for alternative dispute resolution, settlements, and agreements between parties (including families), and these cases may not be included in the data provided by the Supreme Court. Unfortunately, not all cases resolved outside of the court system are made public at this time.

The most significant decrease in the number of closed medical malpractice claims per specialty was observed in OB/GYN, and the probability of a medical professional in OB/GYN being involved in a lawsuit has also decreased. What is striking is the reason for the significant decrease in closed medical malpractice claims in OB/GYN compared to other medical specialties. In the United States of America, OB/GYN departments are positioned as high-risk for litigation, with high rates of litigation and paid claims, including settlements and high compensation claims [18]. As of 2006, OB/GYN had the third highest number of closed medical malpractice claims in Japan after internal medicine and surgery and had the most closed malpractice claims per 1,000 physicians, making it a high-risk department for litigation. However, in 2009, an obstetric care compensation system was established to compensate patients and families for delivery accidents and prevent disputes [19]. Based on our analysis, before the introduction of the obstetric healthcare compensation system in 2009, there were over 100 closed medical malpractice claims in the field of OB/GYN on average each year. However, considering the average trial duration for OB/GYN cases (29.1 months), approximately 3 years

after the system's implementation, the number of closed medical malpractice claims decreased by about half in 2012 (59 cases), and in recent years, it has reduced to approximately one-fourth from the peak period. Furthermore, the number of closed medical malpractice claims per 1,000 physicians decreased by over 70%, indicating that this compensation system contributed to reducing the risk of litigation for OB/GYN, which is supported by previous literature [20, 21].

On the contrary, there has been a marginal escalation in closed medical malpractice claims associated with plastic surgery in Japan. Correspondingly, there has been a mounting trend of lawsuits concerning plastic surgery in the United States of America, with a significant proportion of plastic surgeons expected to face legal action at least once [22]. A review indicated that most plastic surgery litigation cases in various countries are linked to cosmetic procedures, resulting in lower compensation than other medical disciplines [22–24]. Furthermore, our analysis supports the notion that plastic surgery is considered a relatively high-risk area for litigation in Japan.

More than half of the closed medical malpractice claims in Japan are resolved by settlement. For example, in the United States of America, more than half of the cases are dismissed by the courts, and of those, about 45% proceed to settlements and judgments [25].

In a study comparing the characteristics of paid claims between settlements and judgments in the United States of America, cases with obvious diagnostic or treatment errors and those in which the patient died were often resolved by settlement [8]. The tendency to settle medical lawsuits has been shown to be due to lower compensation payment on average in a settlement than in a judgment since the trial period is shorter in a settlement, and the psychological burden on both sides is also lower [8]. Since published studies of publicly available medical litigation in Japan have been analyzed using only concluded medical litigation cases, detailed characteristics of actual settlement cases and their reasons for settling need further study [9–12, 26, 27]. In this study, plastic surgery and OB/GYN were the medical specialties with the highest percentage of settlements among the medical lawsuits in Japan. In particular, OB/GYN lawsuits are more expensive than other medical specialties in terms of compensation, with the amount of compensation in a judgment being twice the amount of that for a settlement [28], and a patient claim is more likely to be approved in favor of the patient in a judgment than in other medical specialties [25]. This indicates that there is a propensity to settle a case through settlement before the conclusion of the trial to reduce the high amount of compensation in the judgment. Conversely, closed medical malpractice claims in psychiatry have significantly lower settlement and acceptance rates than those in other medical specialties. The reason for this cannot be clarified in the present study. However, studies in the United States of America and other countries have demonstrated that psychiatric lawsuits are slightly increasing every year, and the reason may be that some cases develop into lawsuits regardless of social or medical safety issues [29–31]. To enhance patient safety and elevate healthcare quality through insights gained from medical malpractice claims, it will be increasingly crucial to foster improved communication between healthcare providers and patients in the future, with a dedicated emphasis on engaging patients actively [32]. Also, in order to decrease the occurrence of distressing medical errors that result in legal actions, healthcare organizations must grasp both patients' and healthcare professionals' viewpoints. Consequently, medical professionals can establish patient safety management protocols to delve into the causes of medical errors, implement corrective measures, and cultivate a culture of safety.

This study has some limitations. First, our dataset encompasses information on all closed medical malpractice claims in Japan; however, it does not include specific details such as the number of judgments or the identities of the defendants (whether a doctor or a hospital). Nevertheless, limited prior research has explored litigation trends or characteristics in Japan, and

the present study will add strong evidence of the malpractice claims among each specialty in Japan. Second, Japan's cultural background and judicial system limit the number of lawsuits, making it impossible to determine litigation risk and unfeasible to compare litigation trends with other countries. It is noteworthy that, at present, the data on medical malpractice cases and the Supreme Court's involvement in Japan cannot be quantified as in numerous US medical litigation studies [4, 6–8]. However, through the comprehensive investigation in this study, the actual situation of closed medical malpractice claims by medical specialty has been revealed in Japan.

## Conclusion

This study of 13,340 closed malpractice claims in Japan over 16 years revealed that half of the cases were settled, and the percentage of cases won by the patients was low. Notably, closed medical malpractice claims in Japan have decreased in most medical specialties since 2006.

## Acknowledgments

Acknowledgments

We express our appreciation to the Public Relations Section, Supreme Court of Japan, for their kind assistance in extracting the medical malpractice claim data. We also thank Mr. Shohei Mituhashi J.D. for his careful advice as a lawyer and for sharing his pearls of wisdom with us during this research.

## Author Contributions

**Conceptualization:** Kaori Taniguchi, Takashi Watari, Kiwamu Nagoshi.

**Data curation:** Kaori Taniguchi, Takashi Watari, Kiwamu Nagoshi.

**Formal analysis:** Kaori Taniguchi, Kiwamu Nagoshi.

**Investigation:** Kaori Taniguchi, Kiwamu Nagoshi.

**Project administration:** Kiwamu Nagoshi.

**Supervision:** Takashi Watari, Kiwamu Nagoshi.

**Validation:** Kiwamu Nagoshi.

**Writing – original draft:** Kaori Taniguchi.

**Writing – review & editing:** Kaori Taniguchi.

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
