## [Decision Letter · Decision Letter 0]

21 Jun 2023

PONE-D-23-13818Characteristics and trends of medical malpractice claims in Japan between 2006 and 2021PLOS ONE

Dear Dr. NAGOSHI,

Thank you for submitting your manuscript to PLOS ONE. After careful consideration, we feel that it has merit but does not fully meet PLOS ONE’s publication criteria as it currently stands. Therefore, we invite you to submit a revised version of the manuscript that addresses the points raised during the review process.

We look forward to receiving your revised manuscript.

Kind regards,

Alessandro Vittori, M.D.

Academic Editor

PLOS ONE

Journal Requirements:

Reviewers' comments:

Reviewer's Responses to Questions

**Comments to the Author**

1. Is the manuscript technically sound, and do the data support the conclusions?

Reviewer #1: Yes

Reviewer #2: Yes

Reviewer #3: Partly

2. Has the statistical analysis been performed appropriately and rigorously? 

Reviewer #1: Yes

Reviewer #2: Yes

Reviewer #3: No

3. Have the authors made all data underlying the findings in their manuscript fully available?

Reviewer #1: Yes

Reviewer #2: Yes

Reviewer #3: Yes

4. Is the manuscript presented in an intelligible fashion and written in standard English?

Reviewer #1: Yes

Reviewer #2: Yes

Reviewer #3: Yes

5. Review Comments to the Author

Reviewer #1: The authors have put together a concise and organized manuscript on litigation of medical malpractice in Japan. Similar analyses have been done in other jurisdictions but each legal and medical system can have different trends worth exploring as they have done here.

For future study it may be worth examining characteristics of successful cases (patient harm, grievousness of medical error, etc)

Accept with minor edits discussed below

Line 124: appears to be erroneous “s” in line

Line 164 – becomes double spaced.

Fig3 – Can you clarify if this included only those trials that went to judgement and excluded settlement? Either way please clarify how length of trial was measured as it’s not mentioned prior to discussion of this figure

Line 255 – May be worthwhile highlighting Ob/gyn litigation before and after the 2009 change (rather than just expressing a decrease between 2006 and 2021 overall)

Line 451 – Supplementary figure S1 is more core to the manuscript than current figure 3. Suggest swap

Reviewer #2: Thank you for the opportunity to review this work. A few comments:

- This is an observational/descriptive study. Please be careful of using language that may be misleading in terms of causation (e.g. in the abstract line 59, "As a result..." - in reality the study does not identify the reason for the decrease, but the study does identify a decrease...)

- I am not sure that I would call medical malpractice the "root cause analysis" of medical errors for training and improvement. Really, quality improvement, root cause analyses and morbidity/mortality conferences should be conducted locally and improvements made BEFORE medical malpractice lawsuits develop. I would change this throughout the manuscript.

- Do all medical malpractice cases result in supreme court evaluation/appearance? How do the commercial databases differ from the supreme court database?

- In the methods, what does " fully bibliographic" mean? Is this data deidentified? Please clarify.

- Lines 169-170: I'm not sure the data say that patient "demands" were rejected three times more than those of medical practitioners. The sentence and awkward and needs to be rewritten.

- Table one - please include percentages in each cell

- Figure one - difficult to interpret without having the number of cases standardized to something. I'm not sure this figure really adds much and probably can or should be removed.

- Can you identify from this data any reason that the rates of medical malpractice cases per 1000 physicians are decreasing over time? Has the court been finding in favor of the physicians over time?

- How does the Supreme court limit the number of medical malpractice cases? Please elaborate.

- Did the data contain any information as to the allegations in the malpractice suits and why the suits were filed? How much the settlements were?

- A test of trend may be useful to evaluate the trends in table 2 and whether they are important. For example, the rates for plastic surgery appear to be stable, but one can also interpret this as an increase. A statistical assessment here (e.g. Chi-square for trend of Cochrane-Armitage test) would be helpful.

Reviewer #3: The authors present the changing trends in medical litigation in Japan.

1. The authors may clarify whether Others that are not included in Judgments, settlements, or withdrawals.

2. The author simply described the change. It would be better to present a statistical analysis, like that the change was more pronounced in obstetrics and gynecology than in other specialties.

3. Is there any information that is relevant to the lawsuit other than the specialty? For example, the amount of awarded by the judgment, the defendant party (doctor or hospital), etc.

4. Authors may want to add similar studies in Japan or other countries besides the US and discuss of them.

5. Overall, the description of the statistics is poor, and the readability of the figure is low. Current figures simply show the average value and does not provide any additional information. You may need to improve readability (Figure 1, 2)or include more details (Figure 3).

6. PLOS authors have the option to publish the peer review history of their article (what does this mean?). If published, this will include your full peer review and any attached files.

Reviewer #1: No

Reviewer #2: **Yes: **Issam Koleilat

Reviewer #3: No

---

## [Author Response · Author response to Decision Letter 0]

25 Aug 2023

We carefully studied the reviewers’ feedback and revised the manuscript accordingly. Please find below our comprehensive point-by-point responses to the reviewers' comments. 

Reviewer 1

The authors have put together a concise and organized manuscript on litigation of medical malpractice in Japan. Similar analyses have been done in other jurisdictions but each legal and medical system can have different trends worth exploring as they have done here. For future study it may be worth examining characteristics of successful cases (patient harm, grievousness of medical error, etc) 

Accept with minor edits discussed below

1. Line 124: appears to be erroneous “s” in line

Response 1: 

Thank you for bringing this to our attention. We apologize for the oversight, and have removed the "s" in Line 132 (formally Line 124). 

2. Line 164 – becomes double spaced.

Response 2: 

Thank you for pointing out this error. We appreciate your feedback, and have made the necessary correction in the main text in Lines 199–245 (formally Line 164–217); the lines are double-spaced.

3. Fig3 – Can you clarify if this included only those trials that went to judgement and excluded settlement? Either way please clarify how length of trial was measured as it’s not mentioned prior to discussion of this figure

Response 3:

Thank you for your important question. We apologize for the insufficient explanation. The trial duration encompasses not only the judgment but also includes settlements and all other legal proceedings. Therefore, we have added the following to the main text and caption of the figure:

Changes: Definitions of the claim cases Lines 167–169

“The trial duration refers to the period from when the court accepted the lawsuit until a case concluded with a judgment, settlement, withdrawal, or any other legal resolution.”

4. Line 255 – May be worthwhile highlighting Ob/gyn litigation before and after the 2009 change (rather than just expressing a decrease between 2006 and 2021 overall)

Response 4:

Thank you for the valuable feedback. Previous literature on the qualitative research of the obstetric healthcare system also suggests that this system contributes to a reduction in litigation [22]. Our data also clearly showed a decrease in litigation since 2009. Therefore, as you rightly pointed out, it is essential to emphasize this point. Accordingly, we revised the main text as follows:

Changes: Discussion Lines 294–300 

“Based on our analysis, before the introduction of the obstetric healthcare compensation system in 2009, there were over 100 closed medical malpractice claims in the field of OB/GYN on average each year. However, considering the average trial duration for OB/GYN cases (29.1 months), approximately 3 years after the system's implementation the number of closed medical malpractice claims decreased by approximately half in 2012 (59 cases), and in recent years, it has reduced to approximately one-fourth from the peak period.”

5. Line 451 – Supplementary figure S1 is more core to the manuscript than current figure 3. Suggest swap

Response 5:

Thank you for your kind suggestion. Supporting information Fig S1 graphically represents Table 1; however we added it as a supplement to visually emphasize the differences among medical departments. We also agree that FigS1 is important to the core of the manuscript, and we refer to it as Fig1. Therefore, we have modified the notation of the figure as follows.

Changes: Results Lines 215-216

“The distribution of closed medical malpractice claims outcomes by medical specialty is further illustrated in Fig 1.”

Changes: Results Line 218 figure 1 (formally supplementary figure S1)

Changes: Results Line 230 figure 2 (formally figure 1)

Changes: Results Line 239 figure 3 (formally figure 2)

Changes: Results Line 247 figure 4 (formally figure 3)

Reviewer 2 

Thank you for the opportunity to review this work. A few comments:

1. This is an observational/descriptive study. Please be careful of using language that may be misleading in terms of causation (e.g. in the abstract line 59, "As a result..." - in reality the study does not identify the reason for the decrease, but the study does identify a decrease...)

Response 1:

Thank you for your valuable feedback. We apologize for the misleading description that might have caused a misinterpretation of the data, despite it being a descriptive study. Accordingly, we have made the following revisions in the Abstract: 

Change: Abstract Lines 59-63 (formally Lines 57-60)

“By contrast, psychiatry cases exhibited a lower likelihood of settlement, and the percentage of cases resulting in unfavorable outcomes for patients was notably high. Furthermore, there has been a decline in the number of closed medical malpractice claims in Japan in recent years compared to the figures observed in 2006.”

2. I am not sure that I would call medical malpractice the "root cause analysis" of medical errors for training and improvement. Really, quality improvement, root cause analyses and morbidity/mortality conferences should be conducted locally and improvements made BEFORE medical malpractice lawsuits develop. I would change this throughout the manuscript. 

Response 2:

Thank you for your valuable suggestion. Indeed, the expression "positioning medical litigation as 'root cause analysis' or 'educational tool' " was too presumptuous. What we meant to convey is that through categorizing and analyzing numerous cases presented by past medical malpractice litigations, they can serve as useful subjects of analyses for systematically considering healthcare safety measures. Thus, with this in mind, we revised the following statement accordingly:

Change: Abstract Lines 46-48: 

Classification and analysis of existing data on medical malpractice lawsuits is useful in identifying the root causes of medical errors and in considering measures to prevent recurrence.

Change: Introduction Lines 102-108: 

“Ideally, it is desirable to present incidence and mortality rates resulting from medical errors and conduct a root cause analysis before cases escalate into medical litigation. However, categorizing and analyzing existing data from medical malpractice litigations can also be valuable for identifying the root causes of medical errors and considering measures to prevent their recurrence. This process has the potential to contribute to the improvement of the next generation of healthcare, ensuring higher quality standards.”

3. Do all medical malpractice cases result in supreme court evaluation/appearance? How do the commercial databases differ from the supreme court database?

Response 3: 

Thank you for your important questions. Not all medical malpractice cases are resolved at the Supreme Court level. The data held by the Supreme Court includes the total number of resolved medical malpractice cases, including those resolved at lower courts, along with their respective outcomes, such as judgments, settlements, dismissals, and other classifications. These cases were concluded or resolved through various legal procedures. In contrast, the precedent database in our country only contains judgment cases, excluding results such as settlements.

4. In the methods, what does " fully bibliographic" mean? Is this data deidentified? Please clarify.

Response 4:

We apologize for any misunderstandings arising from the ambiguity inherent in our choice of wording. Our dataset encompasses information pertaining to all domestic medical claims in Japan. In this context, we employed the term "fully bibliographic" to convey our comprehensive knowledge of all the cases within Japan. However, it is important to note that this does not imply that we possess exhaustive medical details for each individual case. Hence, modifications have been implemented as follows:

Change: Ethics statement Lines 172-176 (Formally Line 148-151)

“Since the data used in this study are publicly available and the statistical data obtained from the Supreme Court do not include comprehensive medical details of individual cases, no approval was required by the Institutional Review Board of Shimane University, and informed consent was not applicable.”

5. Lines 169-170: I'm not sure the data say that patient "demands" were rejected three times more than those of medical practitioners. The sentence and awkward and needs to be rewritten.

Response 5: 

Thank you for your understanding and recommendation. You are correct that the number of cases varies depending on the medical specialty, and our previous statement may have been misleading. We revised the sentence as follows:

Change: Results Lines 194-197 (Formally Lines 167-170)

“Regarding judgments, out of a total of 4,734 cases (35.3%), 3,589 cases (75.8%) resulted in rejected claims and 1,145 cases (24.2%) resulted in accepted claims. This indicates a higher frequency of patient claims were dismissed than judgment claims against practicing physicians.”

6. Table one - please include percentages in each cell

Response 6:

Thank you for your valuable suggestion. We added the percentage values to the "Acceptance rate (%)" column in Table 1.

7. Figure one - difficult to interpret without having the number of cases standardized to something. I'm not sure this figure really adds much and probably can or should be removed.

Response 7:

Thank you for your valuable feedback. As you rightly pointed out, since the case numbers are not standardized, direct comparisons between medical specialties are not feasible. However, we believe it is still possible to explain the trends in the number of litigations for each medical specialty by visualizing the data over time. Fig 1 allows us to illustrate which specialties have had a decrease or increase in litigations.

8. Can you identify from this data any reason that the rates of medical malpractice cases per 1000 physicians are decreasing over time? Has the court been finding in favor of the physicians over time?

Response 8:

Thank you for your valuable insight. Based on the data at hand, we can only observe the numerical decrease in the occurrence rate of medical malpractice incidents per 1000 physicians over time; we cannot identify specific scientific evidence or describe further details beyond these numerical trends.

Regarding the reasons the courts have been rendering more favorable judgments over time, our research was limited to numerical information, making it difficult to establish causal relationships. However, it can be speculated, based on previous literature, that around 2006, during the period when medical litigation was more heated, judgments may have been somewhat more stringent towards healthcare providers than they are today. High transparency in the social context may have contributed to the adoption of a broader concept of informed consent, leading to stricter judgments against deficiencies in informed consent.

Change: Discussion Lines 269-276:

“As for the reasons courts are ruling in favor over a long period of time, the study was limited to numerical information, and it was difficult to establish a causal relationship. However, prior literature suggests that high transparency in social contexts may have contributed to the adoption of a broader concept of informed consent and led to stricter judgments about deficiencies in informed consent [17]. We can speculate that around 2006, when medical litigation was substantially increased, the sentences against healthcare professionals were somewhat stricter than they are today.”

9. How does the Supreme court limit the number of medical malpractice cases? Please elaborate.

Response 9:

Thank you for your valuable question and comment. 

The data used in our study strictly pertains to cases filed in the court system. We appreciate your understanding of these limitations and considerations in interpreting the results. We have added the following content to the manuscript as an explanation:

Change: Discussion Lines 277-282

“As far as we know, there is no evidence to suggest that the Supreme Court limits the number of medical malpractice cases. However, it is important to note that Japan’s legal system allows for alternative dispute resolution, settlements, and agreements between parties (including families) and these cases may not be included in the data provided by the Supreme Court. Unfortunately, not all cases resolved outside of the court system are made public at this time.”

10. Did the data contain any information as to the allegations in the malpractice suits and why the suits were filed? How much the settlements were?

Response 10:

Thank you for your important questions. We have added content, quoted below in this response, in the Definitions of the claim cases subsection, Materials and Methods section, of the manuscript for explanation. We hope you understand the current situation regarding the limited availability of settlement-related data and apply these considerations in interpreting the findings. 

Change: Definitions of the claim cases Lines 140-144

“The data provided in this study consists of aggregated information on the number of medical malpractice claims resolved through legal procedures along with the litigation outcomes and average trial durations, categorized by medical specialty and year. It does not include specific details such as the reasons for filing the lawsuits, litigation content, or the settlement amounts.”

11. A test of trend may be useful to evaluate the trends in table 2 and whether they are important. For example, the rates for plastic surgery appear to be stable, but one can also interpret this as an increase. A statistical assessment here (e.g. Chi-square for trend of Cochrane-Armitage test) would be helpful.

Response 11:

Thank you for your valuable suggestion. We believe that using statistical analysis would allow us to interpret the numerical changes accurately. However, we considered to avoid bias due to testing, we chose a descriptive research design form the beginning. In the future, as we pursue trends and causal relationships related to medical malpractice, we will regard the importance of statistical evaluations and gather sufficient data for analyses.

Change: Study design Lines 128-133 (Formally Lines 122-125)

“We included all closed medical malpractice claims cases in Japan from 2006 to 2021. A descriptive research design was chosen for this study due to the limited number of data points and to avoid bias due to testing. In June 2022, we requested and received permission from the Public Affairs Division of the Supreme Court Administrative Office to provide medical malpractice statistics and to publish the data as "Medical Malpractice Claims Case Statistics."

Reviewer 3

The authors present the changing trends in medical litigation in Japan.

1. The authors may clarify whether Others that are not included in Judgments, settlements, or withdrawals.

Response 1:

Thank you for your valuable feedback. As you correctly pointed out, it is necessary to provide further explanation regarding the contents of the "others" category. Therefore, the following addition was made to the main text:

Change: Definitions of the claim cases Lines 145–152

“The results of the closed medical malpractice claims provided by the Supreme Court were categorized as "Judgment," "Settlement," "Medical provider admits patient's claim," "Patient waives own claim," "Withdrawal," and "Other." Of these, "Medical provider approves patient's claim" and " Patient waives own claim" only applied to a few cases and were judged to be a small portion of the total number of cases surveyed; therefore, we included them in the "Other" category. We categorized closed medical malpractice claims into four categories: (a) Judgments, (b) Settlements, (c)Withdrawals, and (d) Others.”

2. The author simply described the change. It would be better to present a statistical analysis, like that the change was more pronounced in obstetrics and gynecology than in other specialties. 

Response 2:

Thank you for your valuable suggestion. We thoroughly comprehend your perspective. The data we possess provide comprehensive and precise accounts of the total number of medical lawsuits within the country. However, they do not inherently include information regarding the causes of these lawsuits, relevant factors involved, or the specific characteristics of the defendants or medical organizations. Consequently, it was not feasible to subject the data to rigorous statistical analyses including a time series analysis or hypothesis testing, such as an interrupted time series (ITS) or difference-in-differences (DID) analysis. Moreover, the study data were not designed to be adjusted for numerous complex confounding factors, such as cultural and political contexts, which are discussed in the limitations paragraph of the Discussion section.

3. Is there any information that is relevant to the lawsuit other than the specialty? For example, the amount of awarded by the judgment, the defendant party (doctor or hospital), etc. 

Response 3:

Thank you for expressing your interest in our research. We sincerely appreciate your question. We have duly acknowledged this limitation in our study. Our primary objective was to offer a comprehensive overview of the cases within the defined parameters. Nonetheless, we recognize the potential value of investigating these specific details for future research endeavors. We extend our gratitude to you for highlighting this aspect. For better clarity we have revised the following content in the limitation paragraph of the Discussion section:

Change: Discussion Lines 339–342

“First, our dataset encompasses information on all closed medical malpractice claims in Japan; however, it does not include specific details such as the amount of the judgments or the identities of the defendants (whether a doctor or a hospital).”

4. Authors may want to add similar studies in Japan or other countries besides the US and discuss of them. 

Response 4:

Thank you for your valuable suggestion. We have made every effort to investigate similar studies in countries other than the United States of America. However, the differences in healthcare systems and legal frameworks make it impossible to make straightforward comparisons, and this has been mentioned as a limitation of our research in the limitations paragraph of the Discussion section.

5. Overall, the description of the statistics is poor, and the readability of the figure is low. Current figures simply show the average value and does not provide any additional information. You may need to improve readability (Figure 1, 2)or include more details (Figure 3).

Response 5:

We appreciate your feedback. It seems that our choice of wording may not have been as effective as intended, leading to a potential misunderstanding regarding the level of detail included in our case information, particularly in terms of medical data. While our dataset boasts strength in terms of comprehensive coverage of medical cases in the country, it is important to acknowledge that we do not possess detailed information for every individual case. It is worth noting that the Japanese Supreme Court and medical litigation case data are not currently quantifiable to the extent of CRICO at Harvard University in the United States of America, for example. This limitation represents our greatest weakness. We have duly acknowledged it in the study's limitations paragraph in the Discussion section and added addition content as follows:

Change: Discussion Lines 346–350 

“It is noteworthy that, at present, the data on medical malpractice cases and the Supreme Court's involvement in Japan cannot be quantified as in numerous US medical litigation studies [4,6,7,8]. However, through the comprehensive investigation in this study, the actual situation of closed medical malpractice claims by medical specialty has been revealed in Japan.”

---

## [Decision Letter · Decision Letter 1]

23 Oct 2023

PONE-D-23-13818R1Characteristics and trends of medical malpractice claims in Japan between 2006 and 2021PLOS ONE

Dear Dr. NAGOSHI,

Thank you for submitting your manuscript to PLOS ONE. After careful consideration, we feel that it has merit but does not fully meet PLOS ONE’s publication criteria as it currently stands. Therefore, we invite you to submit a revised version of the manuscript that addresses the points raised during the review process.

The article is almost ready for publication, only a minor revision is needed to complete the optimization of the manuscript (see reviewer 3 and 4's comments). 

We look forward to receiving your revised manuscript.

Kind regards,

Andrea Cioffi

Academic Editor

PLOS ONE

Journal Requirements:

Reviewers' comments:

Reviewer's Responses to Questions

**Comments to the Author**

1. If the authors have adequately addressed your comments raised in a previous round of review and you feel that this manuscript is now acceptable for publication, you may indicate that here to bypass the “Comments to the Author” section, enter your conflict of interest statement in the “Confidential to Editor” section, and submit your "Accept" recommendation.

Reviewer #1: All comments have been addressed

Reviewer #2: All comments have been addressed

Reviewer #3: All comments have been addressed

Reviewer #4: All comments have been addressed

2. Is the manuscript technically sound, and do the data support the conclusions?

Reviewer #1: Yes

Reviewer #2: Yes

Reviewer #3: Yes

Reviewer #4: Yes

3. Has the statistical analysis been performed appropriately and rigorously? 

Reviewer #1: Yes

Reviewer #2: Yes

Reviewer #3: Yes

Reviewer #4: Yes

4. Have the authors made all data underlying the findings in their manuscript fully available?

Reviewer #1: No

Reviewer #2: Yes

Reviewer #3: Yes

Reviewer #4: Yes

5. Is the manuscript presented in an intelligible fashion and written in standard English?

Reviewer #1: Yes

Reviewer #2: Yes

Reviewer #3: Yes

Reviewer #4: Yes

6. Review Comments to the Author

Reviewer #1: All comments addressed. Well worded response to prior. Changes to other reviewer comments also made. Recommend acceptance for publication.

Reviewer #2: Thank you for allowing me to review this revision. The authors appear to have addressed my comments.

Reviewer #3: The authors replied to the reviewer's points properly except one point; the figures still have readability issues. For line graphs, you can put the shape of the line (e.g. dotted line or short-dashed line) as well as the color, or for box graphs, you can fill the box with a pattern.

Reviewer #4: This study contains valuable information. In past studies, the frequency of medical judicial complaints has been investigated, but the trend of this frequency has been less studied.

The following suggestions are recommended for this research:

1- In the discussion section, it should be mentioned the preventive application of these judicial complaints in hospital places, etc.

2- Will the decrease in the frequency of medical lawsuits over time mean a decrease in the frequency of complaints and dissatisfaction of patients in hospitals and clinics, etc.? This topic will be addressed in the discussion section.

7. PLOS authors have the option to publish the peer review history of their article (what does this mean?). If published, this will include your full peer review and any attached files.

Reviewer #1: No

Reviewer #2: **Yes: **Issam Koleilat

Reviewer #3: No

Reviewer #4: No

---

## [Author Response · Author response to Decision Letter 1]

5 Dec 2023

Reviewer #3: The authors replied to the reviewer's points properly except one point; the figures still have readability issues. For line graphs, you can put the shape of the line (e.g. dotted line or short-dashed line) as well as the color, or for box graphs, you can fill the box with a pattern.

Response 

Thank you very much for your valuable feedback. In response to your comments, we have made changes to the line shapes for line graphs, such as dotted lines and short dashed lines. In addition, we have improved the visibility of box graphs by adding a distinctive pattern in the box. We are confident that the clarity of the charts can be communicated more effectively. 

Changes

The box graphs in Figures 1 and 2 were changed to a pattern fill. The line graph in Figure 3 has been changed to a dotted line or a short-dashed line.　Figure 4 is also in black and white for consistency with the other graphs. Further, titles in Figures 2 and 3 have been deleted.

Reviewer #4: This study contains valuable information. In past studies, the frequency of

medical judicial complaints has been investigated, but the trend of this frequency has been less studied. The following suggestions are recommended for　this research:

1- In the discussion section, it should be mentioned the preventive application of these judicial complaints in hospital places, etc.

Response

Thank you for providing valuable and significant feedback. As you have suggested, and in addition to implementing general measures to reduce medical litigation, we have incorporated the following insights from the discussion on patient engagement. The input from the reviewers has played a crucial role in enhancing our paper, and we are genuinely grateful for their contributions.

Changes

Lines 341–349

“To enhance patient safety and elevate healthcare quality through insights gained from medical malpractice claims, it will be increasingly crucial to foster improved communication between healthcare providers and patients in the future, with a dedicated emphasis on engaging patients actively [32]. Also, in order to decrease the occurrence of distressing medical errors that result in legal actions, healthcare organizations must grasp both patients' and healthcare professionals' viewpoints. Consequently, medical professionals can establish patient safety management protocols to delve into the causes of medical errors, implement corrective measures, and cultivate a culture of safety.”

Reviewer #4: 

2-Will the decrease in the frequency of medical lawsuits over time mean a decrease in the frequency of complaints and dissatisfaction of patients in hospitals and clinics, etc.? This topic will be addressed in the discussion section.

Response 

Thank you for your insightful comments. The decrease in the frequency of medical lawsuits does not mean that the frequency of patient dissatisfaction with medical care is decreasing. However, the Medical ADR was established in 2007, and now there are 11 locations nationwide. This has allowed disputes to be handled outside of court, which may have reduced the number of cases leading to litigation. Therefore, the following modifications have been made.

Changes

Lines　267–273S；

Closed medical malpractice claims during the study period peaked in 2006 and have been fluctuating between 700–800 cases annually in recent years. However, the decrease in the number of medical lawsuits probably does not translate into a decrease in the frequency of patient complaints and dissatisfaction with medical care. Medical ADRs were established in 2007 and are now located in 11 locations nationwide; thus, the number of cases leading to lawsuits may have been suppressed as disputes and are now handled outside of court.

---

## [Editor Report · Decision Letter 2]

7 Dec 2023

Characteristics and trends of medical malpractice claims in Japan between 2006 and 2021

PONE-D-23-13818R2

Dear Dr. NAGOSHI,

We’re pleased to inform you that your manuscript has been judged scientifically suitable for publication and will be formally accepted for publication once it meets all outstanding technical requirements.

Kind regards,

Andrea Cioffi

Academic Editor

PLOS ONE

Additional Editor Comments (optional):

No further revisions are necessary.
---

## [Editor Report · Acceptance letter]

10 Dec 2023

PONE-D-23-13818R2 

Characteristics and trends of medical malpractice claims in Japan between 2006 and 2021 

Dear Dr. Nagoshi:

I'm pleased to inform you that your manuscript has been deemed suitable for publication in PLOS ONE. Congratulations! Your manuscript is now with our production department. 

Kind regards, 

on behalf of

Dr. Andrea Cioffi 

Academic Editor

PLOS ONE